# The Effect of Cushioned Centrifugation, with and without Enzymatic Reduction of Viscosity, on the Motility Pattern and Kinematic Parameters of Dromedary Camel Bull Spermatozoa

**DOI:** 10.3390/ani13172685

**Published:** 2023-08-22

**Authors:** Davide Monaco, Giovanni Michele Lacalandra, Zeeshan Ansar, Paolo Trerotoli, Brendan Patrick Mulligan, Taher Kamal Osman

**Affiliations:** 1Department of Veterinary Medicine, University of Bari Aldo Moro, 70010 Valenzano (BA), Italy; 2Department of Advanced Biotechnology and Research, Salam Veterinary Group, Buraydah 51911, Saudi Arabia; 3Interdisciplinary Department of Medicine, University of Bari Aldo Moro, 70124 Bari, Italy; 4Camel Reproduction Centre, Dubai P.O. Box 79914, United Arab Emirates

**Keywords:** epididymal spermatozoa, seminal plasma, ejaculates, dilution, recovery rate, Papain, progressive spermatozoa, curvilinear velocity, average path velocity, agglutination

## Abstract

**Simple Summary:**

Semen processing procedures in the dromedary camel species (*Camelus dromedarius*) are still underdeveloped as compared to other domestic species. The viscosity of the camelid’s ejaculate presents challenges that must be overcome with specialized procedures. In this study, we have proven that seminal plasma affects the motility of dromedary camel epididymal spermatozoa, and we have evaluated the effect of a cushioned centrifugation procedure (900× *g* for 20 min) with and without enzymatic reduction of viscosity for processing dromedary camel ejaculates. The obtained results showed that the dilution of dromedary camel ejaculate and centrifugation with cushion fluid improves the motility pattern of dromedary camel spermatozoa. Moreover, the use of the protease enzyme Papain could help in further reducing the viscosity of camelid ejaculates and in obtaining higher sperm recovery rates following cushioned centrifugation.

**Abstract:**

In order to contribute to the development of semen processing procedures in camelids, the aims of the present study were to evaluate (i) the effect of 35% seminal plasma incubation on dromedary camel epididymal sperm motility and kinematic parameters, (ii) the effects of centrifugation, with cushion fluid and enzymatic reduction of viscosity (Papain + E64) during ejaculate processing, on the motility and kinematic parameters of dromedary camel ejaculates. The incubation with seminal plasma significantly reduced the percentage of progressively motile spermatozoa as well as the proportion of medium progressive spermatozoa whilst increasing the percentage of non-progressive spermatozoa. The centrifugation procedure improved the sperms’ kinematic parameters, and the highest values were observed for samples centrifugated with cushion fluid. The samples treated with Papain + E64 showed a significant increase in both total and medium progressive spermatozoa, along with a reduction of non-progressive spermatozoa (*p* < 0.05). The results of this investigation show that a simple, cheap, and effective procedure, such as cushioned centrifugation, could improve the motility patterns of dromedary camel spermatozoa; in combination with enzymatic reduction of viscosity, this method leads to the best results in terms of recovery rates and sperms’ kinematic parameters.

## 1. Introduction

During ejaculation, epididymal spermatozoa (ES) blend with secretions of the sexual accessory glands (Seminal Plasma: SP), which, in camelids, form a viscous coagulum [1]. Viscous seminal plasma allows the slow release of spermatozoa into the uterus during the time span between mating and ovulation and contributes to sperm adhesion and storage in the oviduct [1,2].

The effects of the addition of seminal plasma on ES parameters have been investigated in several species [3,4,5,6,7,8], including camelids [9,10]. However, to date, no studies have investigated changes in motility patterns in ES following exposure to SP in dromedary camels.

Although the use of artificial insemination could significantly contribute to camel breeding efforts worldwide, the viscous nature of the camelids’ SP has hindered the development of semen processing procedures [11]. The liquefaction of an ejaculate is a necessary step for effective sperm evaluation and preservation, and different techniques have been developed for this purpose [12,13,14]. In the dromedary camel (DC) species, ejaculates have typically been diluted (1:1, *v*/*v*) with commercial extenders (Green Buffer, Optixcell, Triladyl), followed by incubation in a water bath at 35 °C for 30 min to achieve liquefaction [15,16]. Greater dilutions (1:5 or 1:10 *v*/*v*) with Tris-citrate-fructose buffer, in combination with intermittent gentle pipetting for 45–60 min at room temperature, have also been employed for this purpose [17].

The “sperm washing” (i.e., dilution and centrifugation of ejaculates) is a common procedure used to reduce seminal plasma concentration and/or to increase sperm concentration in extended or over-diluted ejaculates. However, attempting to maximize sperm recovery through high-speed-high force centrifugation could have an adverse effect on spermatozoa [18]. The method of cushioned centrifugation has, therefore, become a popular technique in many species, as it helps maximize sperm harvest by allowing greater centrifugation time and force without apparent injury to spermatozoa. The cushion fluid is a non-ionic iodinated compound, “iodixanol”, which reduces the pressure that the spermatozoa undergo during centrifugation [19]. Whilst colloid centrifugation protocols have been previously described [12,13], to the best of our knowledge, no studies have yet evaluated the effects of a sperm-washing procedure with the use of cushion fluid in camelids.

Amongst the procedures that have been used in order to reduce the viscosity of the camelids ejaculate, the use of the cysteine protease Papain from Carica papaya has been utilized [20,21,22]. In Alpacas, the use of 0.1 mg/mL of Papain significantly reduces the viscosity of ejaculates within 20 min [22], whereas in the dromedary camel, the same dose-time exposure was considered unsatisfactory [20]. Even when Papain successfully reduced the viscosity (thread formation) of ejaculates, the kinematic parameters of DC spermatozoa, following this enzymatic reduction of viscosity, were not investigated.

Considering the aspects discussed above, the present study was aimed at evaluating the effect of seminal plasma on the motility patterns of DC epididymal spermatozoa and also to test a new protocol for processing camelids’ ejaculates. Ejaculates were processed via a cushioned centrifugation procedure, with and without enzymatic reduction of ejaculate viscosity, and the effects, upon DC sperms’ motility and kinematic parameters were assessed.

## 2. Materials and Methods

The study was carried out in Buraida (Saudi Arabia) at Salam Veterinary Group Camel Hospital from February to March 2023. The experimental protocol was evaluated and approved by the ethics committee for animal studies of the Department of Veterinary Medicine of the University of Bari Aldo Moro (Prot. n. 399-III/13, Approval number 5/2022). All procedures were performed in conformity with European Union Directive 2010/63/EU.

### 2.1. Experimental Design

Three separate experiments were conducted. In experiment 1 (Exp 1), epididymal spermatozoa were collected via retrograde flushing from dromedary camel testis; selected samples (concentration of >300 × 10^6^ spz/mL, total motility > of 75%) were pooled, diluted 1:1 with 70% SP, and evaluated after 45 min incubation at 37 °C.

In experiment 2 (Exp 2), dromedary camel ejaculates were collected with an artificial vagina (AV). Selected ejaculates (non-contaminated ejaculates with a concentration >250 × 10^6^ spz/mL, mass motility score ≥ 3) were diluted with a Tris-Citrate-Fructose extender (TCF) and evaluated before and after centrifugation with and without cushion fluid.

In experiment 3 (Exp 3), ejaculates were collected via AV, and selected ejaculates were treated with Papain in order to eliminate the viscosity and then evaluated before and after a cushioned centrifugation.

### 2.2. Experiment 1

#### 2.2.1. Extenders and Seminal Plasma Preparation

The seminal plasma used in Exp 1 was obtained from two adult dromedary bulls of proven fertility and was collected by the AV. The selected ejaculates (no. 4, non-contaminated, volume ≥ 4 mL, sperm concentration > 300 × 10^6^ sperms/mL, and total motility >80%) were supplemented with 30% (*v*/*v*) TCF. Diluted ejaculates were kept at 4 °C, pipetted intermittently, and allowed to liquefy at 4 °C. Following liquefaction, samples were centrifuged twice at 1000× *g* for 30 min at 4 °C or until the recovered supernatant was completely free of spermatozoa. Seminal plasma samples were pooled and stored in aliquots at −80 °C until use. The Tris–citrate–fructose buffer (TCF) was a modification of the extender proposed by Malo et al. (2017) [13] and composed of 300 mM (3.634 g) TRIS; 94.7 mM (1.99 g) citric acid monohydrate; 27.8 mM (0.5 g) fructose; 0.05%, (*wt*/*v*) Bovine serum albumin; 10 mM (0.29244 g) EDTA; 0.25mM (0.00373 g) D-Penicilamine [23]; adjusted for pH 6.9 and with an Osmolarity around 320 mOsm/kg. The buffer was prepared once a week, sterilized by filtration (0.22 μm), and kept at 4 °C until use. 

#### 2.2.2. Epididymal Semen Collection and Evaluation

Adult (>5 years of age) dromedary camel bull testis were retrieved at the local slaughterhouse, kept at room temperature, and processed within two hours of collection. Epididymal Spermatoza were recovered by retrograde flushing with TCF, according to a previously described technique [9,24]. Samples were kept in a water bath at 37 °C and evaluated for concentration (spectrophotometer), morphology (Eosin/Nigrosin staining [25]), and motility. For the evaluation of motility, an aliquot of the sample was diluted 1:10 or 1:20 with TCF containing 3% BSA, reaching a final concentration of 20–40 × 10^6^ spz/mL, and then incubated for 5–10 min at 37 °C. A volume of approximately 4 µL of the semen BSA-TCF solution was then placed in a disposable motility chamber (Goldcyto, Barcellona, Spain), and the motility was evaluated using a Computer Assisted Sperm Analyzer (CASA).

Only samples with a high degree of morphologically normal spermatozoa (>75% including proximal and distal cytoplasmic droplets), a minimal concentration of >300 × 10^6^ spz/mL, and with total and progressive motilities of 75% and >60% were selected. A minimum of three ES samples were pooled in order to avoid individual male/testis confounding factors. After pooling, the control samples (Con) were evaluated for viability (Eosin/Nigrosin), morphology, and motility, then two aliquots of 0.5 mL each were prepared; one aliquot (SP) was diluted 1:1 (*v*/*v*) with prewarmed 70% SP (final concentration 35% SP) whereas the other portion (Con-Time) was diluted 1:1 (*v*/*v*) with TCF. Both samples were kept in a water bath for 45 min at 37 °C and then evaluated again for viability and motility; the latter was evaluated after dilution with 3% BSA-TCF in order to reach the concentration of 20–40 × 10^6^ spz/mL as described before.

#### 2.2.3. Evaluation of the Sperms’ Motility and Kinematic Parameters

The CASA system was set with the following parameters; (i) Magnification × 100; (ii) frame acquisition 45 frames/s; (iii) cut-off values: Drifting 15 µm/s; Static sperms < 10 µm/s; Slow spermatozoa < 30 µm/s; Medium spermatozoa > 30 µm/s < 50 µm/s; Rapid spermatozoa > 50 µm/s; Progressive spermatozoa STR > 70 (Modified from [13]). At least five fields and 500 spermatozoa were evaluated. Total and progressive motility, percentages of rapid progressive, medium progressive, slow and static spermatozoa were calculated in addition to their means of curvilinear velocity (µm/s) (VCL); average path velocity (µm/s) (VAP); straight line velocity (µm/s) (VSL); straightness (%) (STR); linearity (%) (LIN); wobble (%) (WOB); amplitude of lateral head displacement (µm) (ALH); and beat cross frequency (Hz) (BCF). At the end of the experimental session, all the recorded motility sessions (Con, SP, Time-Con) were analyzed by an experienced operator in order to correct possible software evaluation errors (i.e., small particles or cytoplasmic droplets mistaken as spermatozoa, non-motile spermatozoa displaced by the rapid movement of motile spermatozoa, etc.).

### 2.3. Experiment 2

#### 2.3.1. Semen Collection and Evaluation

Semen was collected from 3 fertile DC breeding bulls (*Camelus dromedarius*) (7–13 years old) belonging to the center and housed in single pens of 10 m × 5 m. The males received daily food rations of alfalfa and concentrated mix and had access to water ad libitum.

Ejaculates were collected by using a 40 cm bull artificial vagina (AV) and a female teaser [26]. Once collected, they were transported within 5 min to the laboratory, placed in a water bath at 35–37 °C and evaluated for contamination, sperm concentration, and mass motility: only non-contaminated ejaculates with a concentration > 250 × 10^6^ spz/mL, and a mass motility score ≥ 3 (on a 0 to 5 scale) were selected. 

The selected ejaculates were diluted with TCF between 1:5 to 1:10 in order to reach an approximate concentration of 40–60 × 10^6^ spz/mL and were then kept in a water bath and pipetted with a Pasteur pipette every 3 to 5 min to reduce the viscosity [13].

Twenty minutes after dilution in TCF, the ejaculates (Pre_Centr) were evaluated for Sperm Concentration (improved Neubauer chamber), Viability (Eosin/Nigrosin), plasma membrane integrity (IPM) (Hypo-osmotic swelling test, HOST), and motility. The HOST was performed by diluting 50 µL of the diluted ejaculate with 250 µL of a sodium citrate-fructose solution (75 mOsm/kg); the percentage of reactive sperm (Host+) was evaluated after 30 min of incubation at 37 °C (modified from [27]). A 200 µL aliquot of the diluted ejaculate was further diluted 1:1 with TCF supplemented with 3% BSA (*wt*/*v*) and, after 5 min of incubation at 37 °C, evaluated for motility. 

#### 2.3.2. Semen Centrifugation

After evaluation of motility, two 8 mL aliquots of the TCF diluted samples were placed in 15 mL falcon tubes. Only Samples with a total motility ≥60% were used. One tube was kept as a control (Centr_Con) whilst, in the other one (Centr_Cush), 1 mL of Iodixanol-based cushion medium (Cushion Fluid, Minitube), kept at room temperature, was carefully placed under the extended semen using a 2 mL syringe and a 9 cm spinal needle without the needle rod (Figure 1a). The two aliquots were centrifuged at 900× *g* for 20 min at room temperature with a swing rotor centrifuge (Hermle, Z326K, Wehingen, Germany) [19,28].

After centrifugation, the supernatants were carefully removed, leaving about 1 mL of TCF above the sperm buffer (Figure 1b or above the pellet, and the cushioning fluid of the Centr_Cush sample was carefully aspirated, trying to remove as much as possible without removing any sperm. In both tubes, the remaining sperm-rich solution was again diluted with TCF to the original volume of 8 mL, and the sperm concentration, viability, IPM, motility and kinematic parameters were again evaluated. The sperm recovery rate (%) was calculated as the percentage of recovered spermatozoa according to the total amount of sperm before centrifugation.

#### 2.3.3. Evaluation of the Sperms’ Motility and Kinematic Parameters

The sperm’s kinematic parameters were evaluated with the same settings of Exp 1. At the end of the experimental sessions, all the recorded motility videos (Pre_Centr, Centr_Con, Centr_Cush) were analyzed by an experienced operator for the correction of software evaluation errors. 

### 2.4. Experiment 3

#### 2.4.1. Semen Collection and Evaluation

Ejaculates were collected, evaluated, selected, and diluted as described before; however, in order to avoid possible interaction with the Papain, the TCF extender was prepared without BSA. 

#### 2.4.2. Enzymatic Reduction of Viscosity

Fifteen minutes after the initial dilution and pipetting, the ejaculates were split into two 15 mL tubes. In one tube (Pap), Papain from Carica Papaya (Sigma-Aldrich, St. Louis, MO, USA) was added (final concentration of 0.1 mg/mL [22]), whereas the other tube was kept as a control (Con), TCF was added with the same volume of added Papain. Both the control and Papain-treated samples were pipetted, with the same number of pipetting repetitions (no. = 10) soon after dilution, and twice more (no. = 10 pipetting) in the following 5 min. After 5 min, the Papain reaction was stopped by adding N-(trans-Epoxysuccinyl)-l-leucine 4-guanidinobutylamide (E-64; Sigma-Aldrich, St Louis, MO, USA) (final concentration 10 µM) and TCF of the same volume of the E64 solution was added to the Con sample [22].

When observed, the sperm head-to-head agglutination was scored from 0 to 5 according to the following criteria: 0-absence of agglutination; 0.5 (5% agglutinated sperm); 1 (10% agglutinated sperm); 1.5 (15%); 2 (20%); 2.5 (25%) 3 (30%) and 4 (40%); 5 (≥50%) (modified from [20]). 

The two samples (Con, Pap) were again evaluated for viability and concentration. Two hundred microliter aliquots were again collected from each tube, diluted with TCF-3% BSA, incubated for 5 min at 37 °C, and evaluated for motility and kinematic parameters as described before.

#### 2.4.3. Semen Centrifugation

Six mL of Con and Pap samples were placed in 15 mL falcon tubes, and after loading 1 mL of cushion fluid below the TCF-sperm solution in both tubes, they were centrifuged at 900× *g* for 20 min at room temperature with a swing rotor centrifuge (Hermle, Z326K, Wehingen, Germany). After centrifugation, the supernatant and the cushioning fluid were removed, then the centrifuged samples (Centr_Con and Centr_Pap) were diluted to reach the initial volume and evaluated for concentration, viability, motility, kinematic parameters, agglutination score. At the end of the sessions, the recorded motility videos (Pap, Con, Centr_Con, Centr_Pap) were analyzed by an experienced operator, and software mistakes were corrected. The total sperm recovery rate (%) was calculated after centrifugation for each treatment (Centr_Con and Centr_Pap).

### 2.5. Statistical Analysis

The Shapiro–Wilks test was applied to verify the Gaussian distribution of data, so they were summarised as mean, and standard deviation and parametric tests were applied. If Gaussian distribution was not valid for the data, they were summarised as median and range or interquartile range, and distribution-free tests were applied as appropriate (Wilcoxon, Kruskal–Wallis, Friedman).

In experiment 1 (Con, Con-time, SP), experiments 2 and 3 (sperm’s motility) were analyzed by a repeated measures ANOVA, and if the null hypothesis was rejected, a pairwise post hoc comparison was applied with p-values adjusted according to Bonferroni.

In experiment 1, the comparison of sperm kinematic parameters among groups (Control, Control Time, SP) was analyzed by a general mixed model, with the subject set as random intercept, and the fixed effects were parameter and experimental approach. The post hoc pairwise comparisons were performed via the Tukey test. The same approach by a general mixed model was applied in experiments 2 and 3 (viability), taking into account the intra-subject correlation of the values of the three treatments (Pre_Centr, Centr_ Con, Centr_Cush) in experiment 2 and of the four treatments (Con, Pap, Centr_Con, Centr_Pap) in Exp 3. The post hoc pairwise comparisons were performed via the Tukey test.

In experiment 3, the agglutination data were analyzed using the Friedman test because data did not approach Gaussian distribution. 

Data about recovery rates have been summarised as median and range. Since concentrations were repeated measurements on different aliquots obtained from the same experimental unit, values were transformed into ranks according to the procedure adopted for the Friedman test, and the comparison was performed applying a generalized model in which the dependent variable was represented by the ranks. Sphericity was assessed by means of the Mauchly test, and in case of rejection of the sphericity hypothesis, the *p*-value was corrected according to Huyn and Feldt. Pairwise comparisons were carried out using non-parametric methods (Friedman test, Wilcoxon test for paired data), and the *p*-value was corrected according to Bonferroni.

For all tests, the statistical significance was set at *p* < 0.05. The analysis was carried out by SAS software (SAS Institute Inc., Cary, NC, USA).

## 3. Results

### 3.1. Experiment 1: Effect of SP Incubation on Motility and Kinematic Parameters of DC Epididymal Spermatozoa

Seven pooled epididymal semen samples were used in Exp 1. The effect of SP incubation on ES motility parameters is reported in Table 1. The seminal plasma significantly reduced the percentage of progressively motile spermatozoa as well as the proportion of medium progressive spermatozoa whilst increasing the percentage of non-progressive spermatozoa in comparison to both pre- and post-incubation samples between which no statistically significant differences were found (Table 1). Incubation with seminal plasma induced significant changes in ES kinematic parameters: VCL, VAP, VSL, and ALH (Appendix A). No statistical differences were found between the Control samples and the Control-time samples. On the other side, the motile sperm’s VCL was significantly reduced in SP samples (82.13 ± 8.24) as compared to the Control (122.47 ± 12.16 *p* < 0.05) and to Control-Time (115.83 ± 11.90) samples (*p* < 0.001). Similar statistical differences were also observed for the motile sperm: VAP (SP: 35.56 ± 3.34, Control: 54.16 ± 5.41, Control Time: 49.31 ± 4.71), VSL (SP: 15.22 ± 1.71, Control: 23.03 ± 2.10, Control Time: 22.03 ± 2.56) and ALH (SP: 2.33 ± 0.21, Control: 3.26 ± 0.3, Control Time: 3.12 ± 0.30) (Appendix A).

### 3.2. Experiment 2: Effects of a Cushioned Centrifugation Procedure on Ejaculates’ Sperm Motility and Kinematic Parameters

Eight semen samples were selected and processed in Exp 2. The effects of centrifugation, with and without cushion fluid, on the percentages of motile spermatozoa are reported in Table 2. No statistical differences were found between Pre_Centr and Centr_Con samples, whereas statistical differences were found between Control (Centr_Con) and cushioned (Centr_Cush) centrifugated samples for percentages of progressive motile, medium progressive, and non-progressive spermatozoa. In both centrifuged samples, the percentages of live and HOST-reacted sperms were significantly lower than in the Pre_Centr samples, but no statistical differences were found between samples centrifuged with and without cushion fluid (Table 2). 

The sperms’ kinematic parameters (VCL, VAP, VSL, ALH, and BCF) increased from Pre_Centr to Centr_Con and to Centr_Cush samples, with the highest values observed for cushioned centrifugated samples (Table 3, Appendix A). Statistical differences were observed between motile and progressive spermatozoa of Pre_Centr and Centr_Cush: VCL, VAP, VSL, ALH, and BCF in addition to motile spermatozoa of Centr_Con and Centr_Cush: VCL, VAP, ALH and BCF (Table 3). The details of the kinematic parameters and the comparison among other motile groups (medium progressive, non-progressive) are reported in Appendix A. 

The recovery rates for normal and cushion fluid centrifugation were 78.2 ± 8.3% and 79.6 ± 13.4%, respectively, and no statistical differences were found between the two procedures (*p* = 0.7896).

### 3.3. Experiment 3: Effects of a Cushioned Centrifugation Procedure, with and without Enzymatic Reduction of the Ejaculates’ Viscosity, on Sperms’ Motility and Kinematic Parameters

Seventeen samples were selected and processed in Exp 3. Dilution and pipetting successfully reduced the viscosity of all samples (no thread formation); however, it was noted that the Papain-treated samples appeared more liquefied than the Control: the pipette drops appeared less dense and more liquid.

The effect of Papain treatment on percentages of motile spermatozoa, before and after centrifugation with cushion, are reported in Table 4. Compared to the Control, Papain-treated samples showed a significant increase in progressively motile spermatozoa and medium progressive spermatozoa along with a reduction of non-progressive spermatozoa (*p* < 0.05). Compared to Con, Centr_Con samples showed a significant reduction of the non-progressive spermatozoa, whereas Centr_Pap samples were statistically different for motile, progressively motile, medium progressive, non-progressive, and immotile spermatozoa (*p* < 0.05). The comparison between Centr_Con and Centr_Pap showed significant differences for motile, progressively motile, medium progressive, and immotile spermatozoa (*p* > 0.05). There was no statistical difference between Papain-treated samples before and after centrifugation (Table 4). 

The sperm viability did not differ between the Control and Papain treatment samples before and after centrifugation (Con vs. Pap *p* = 0.851; Centr_Con vs. Centr_Pap *p* = 0.256). Centrifugation, however, induced a significant decrease in the viable spermatozoa for both samples (Figure 2a). The agglutination score did not significantly change among treatments before and after the centrifugation procedure; however, a slight increase in the agglutination score was noticed in the Papain-treated samples (Figure 2b).

The sperm kinematic parameters were significantly influenced by both the Papain treatment and cushioned centrifugation (Table 5): motile spermatozoa VCL, VAP, VSL, ALH, and BCF were significantly higher in Papain vs. Control group, before and after the centrifugation. The progressive motile VCL, VAP, VSL, and ALH were significantly higher in Pap samples, as compared to Con, before centrifugation, whereas, after centrifugation, only VCL, ALH, and BCF were significantly different between Pap and Con (Table 5). 

The sperm kinematic parameters of Centr_Con samples were significantly improved, as compared to Con, by the cushioned centrifugation procedure and in particular for VCL (motile and progressive motile), VAP (motile), VSL (motile), and ALH (motile and progressive motile). The details of the kinematic parameters of the motile groups (medium progressive, non-progressive) are reported in Appendix A; there were no differences between Papain-treated samples before and after centrifugation. 

The median recovery rate of sperm for the Centr_Pap samples was significantly higher (*p* < 0.0001) than the Centr_Con: 86.37% (Range 74.06–92.84) vs. 78.16% (Range 56.09–90.32), respectively. 

## 4. Discussion

Assisted reproductive technologies in the dromedary camel, and particularly ejaculate processing procedures, are still far from what has been developed in other species. Moreover, the effects of seminal plasma on epididymal spermatozoa and its role in sperm preservation procedures have not been exhaustively investigated. This study aimed to evaluate the effect of supplementation with 35% SP incubation on dromedary camel ES’ motility as well as to evaluate the effect of centrifugation procedures with cushion fluid, with and without enzymatic reduction of viscosity, for the processing of ejaculated spermatozoa. 

Due to the viscous nature of the camelids’ SP, the results of Exp 1 were somewhat expected; however, it should be noted that prior to incubation and motility evaluation, all samples were diluted at least 1:5 with 3% BSA-TCF; therefore, it is unlikely that the observed effect of reduced motility was due to residual SP viscosity. An effect of SP on the sperms’ head adhesion properties, however, cannot be excluded [2]. 

Kershaw–Young and Maxwell (2011) [10] previously evaluated the effect of different percentages of seminal plasma on Alpaca ES. These authors observed that the total motility (progressive and oscillatory, evaluated together and subjectively) decreased from 60% to 55% in ES incubated for 30 min with both 10% and 25% SP and that the motility percentage decreased to 40% if ES were incubated with 50% SP. In the same study, the ES motility was reduced from 60% to 48%, 32%, and 12% in ES samples after 60 min of incubation with 10%, 25%, and 50% SP, respectively. In another experiment, ES motility decreased from 56% to 6%, 43%, or 16% after 1 h incubation with 0, 10%, and 100% SP, respectively [10]. Fumuso et al. (2018) [29] evaluated the effect of SP on ejaculated Alpaca spermatozoa and found that, after 1.5 h incubation, the total sperm motility was reduced from 26% to 17%, 22% and 24% in samples incubated with 0, 10% and 50% SP whereas the progressive sperm motility changed from 16% to 9.1%, to 16% and to 9.3% in samples incubated with 0%, 10% and 50% SP. According to the above-mentioned results, it seems that the effect of incubation with seminal plasma upon camelid ES is conflicting, most probably because of the subjective evaluation of sample motility. 

In the present study, an SP concentration of 35% (*v*/*v*) was investigated only: the addition of TCF was a necessary step for decreasing ejaculate viscosity during storage at 4 °C. Processing of ejaculates at reduced temperatures was performed in order to reduce the metabolic activity of spermatozoa (during liquefaction and centrifugation); this precaution could possibly have prevented the observation of any negative effects resulting from altered seminal plasma. 

In our opinion, the results of the present study cannot be compared with the results obtained by Kershaw–Young and Maxwell (2011) [10]: the significant reduction of the sperms’ motility observed in 0% SP incubated samples could indicate that other factors might have affected their results. For instance, ES were recovered via mincing and underwent washing and centrifugation before incubation. In the present study, ES were collected via retrograde flushing, which allows the recovery of highly concentrated samples (>250 × 10^6^ spz mL) with limited contamination with blood or other materials. In view of the results obtained by other authors regarding collection methodologies for ES [30,31,32], it could be hypothesized that both procedures (mincing and centrifugation) impaired the ES’ functionality, regardless of SP incubation, in the study carried out in Alpacas [10].

The effect of 15% seminal plasma on ES motility before and after freezing has been evaluated previously [9]; the addition of seminal plasma did not cause significant variation in the sperm’s motility before freezing; however, the total and progressive motility were subjectively evaluated and only after 4 °C incubation (pre-freezing evaluation). It is also possible that the 15% SP was not effective in terms of biological activity for impairing the sperm motility pattern, such as that observed in the present study, with 35% SP. However, in view of the wide range of SP percentages evaluated by other authors, the results of the present study may be interpreted cautiously. Further studies are required in order to understand the effect of seminal plasma concentration and its composition in extenders upon the functionality and preservation of dromedary camel spermatozoa. 

Whilst centrifugation for the processing of camelid ejaculates has been used routinely, the current study is the first in which the kinematic parameters of DC sperm were evaluated following a sperm-washing procedure. Morton et al. (2012) [33] diluted alpaca ejaculates, centrifuged them at 600× *g* for 7 min, and obtained a 50% recovery rate in addition to a non-significant improvement in motility (progressive and oscillatory, evaluated together and subjectively from 50.0 ± 5.8% to 66.7 ± 6.7%). El–Bahrawy (2017) [21] centrifuged dromedary camel ejaculates, without cushion fluid, at 700× *g* for 10 min at °C, obtaining recovery rates between 75% and 85%. However, the motility evaluation in this study was performed only after 2 and 4 h of incubation at 4 °C and increased from 65% to 73% 2 h after centrifugation before decreasing to 47% at 4 h. In the same study, the post-thaw effects of several treatments (Amylase, Caffeine) on sperm kinetic parameters were reported, but data about kinematic parameters before and after centrifugation and without any other treatment were not reported.

In the present study, a higher centrifugation regime (900× *g* × 20 min) was used, and the results, in terms of recovery rates, are almost comparable to previously reported results [21]. The control centrifugation group did not significantly differ from the Pre_centr, but centrifugation with Cushion fluid induced a significant increase in medium progressive spermatozoa and a reduction of non-progressive spermatozoa (Table 2). The centrifugation procedure, however, also induced a decrease in sperm viability and membrane permeability in both treatments. This observation is surprising since a difference between the two centrifugation procedures was expected. At the same time, this finding could also indicate that other factors, such as the centrifugation temperature, could have affected those parameters; this aspect requires further investigations, possibly with more advanced techniques (e.g., fluorescence probes: SYBR-14/Propidium iodide [25]). 

The increase in the kinematic values in both centrifuged samples, as observed in the present study (Table 3 and Appendix A), suggests that the dilution of ejaculates, as previously proposed [16,17], is effective in reducing the ejaculates’ “macro-viscosity” (i.e., the thread formation) but a “micro-viscosity” still persists and could be reduced with a “washing-centrifugation” procedure. The higher performance obtained in the current study with cushion fluid is in agreement with observations made by other authors on stallions [19]. The beneficial effects of cushion fluid could be explained by its softening effect and because of the distribution of the spermatozoa on a larger surface, i.e., forming a sperm buffer instead of a pellet. On the contrary, the pellet formed during normal centrifugation could limit the benefits of washing due to the sticking of spermatozoa pressed to the bottom of the tube. Although the present results look promising, different centrifugation regimes (varying centrifugal forces and time) at different temperatures (RT or 4 °C) and with different types of tubes (15 or 40 mL) should be tested in order to find the best processing protocol for dromedary camel ejaculates. 

The current study is the first to report the effect of enzymatic reduction of SP viscosity upon DC sperm motility, as well as the impact of a cushioned centrifugation procedure. The observed data confirms the ability of Papain to reduce the viscosity of DC ejaculates as well as improve recovery rates, the percentage of motile sperm, and their kinematic parameters. 

The exposure of DC ejaculates to Papain (0.1 mg/mL) for 5 min was more effective in achieving viscosity reduction, as compared to the 20–30 min previously reported in dromedary camels [20] and in Alpacas [22]. This observation is probably due to dilution with TCF (1:5 to 1:10) and lets us suppose that lower Papain doses may be effective when samples are already diluted. Similar to the present results, an improvement in motility (around 10%) and kinematic parameters of Llama spermatozoa was also observed following 20 min of Papain incubation and the addition of E-64 [34]. It is worth noting that, in Exp 3, significant differences were found between the kinematic parameters of Con and Centr_Con whilst no differences were found between Pap vs. Centr_Pap (Appendix A), thus confirming (i) the efficacy of the cushioned centrifugation; (ii) the maximal effect obtained by Papain in terms of kinematic parameter improvement; and (iii) the absence of negative effects of the Papain–Cushion centrifugation on kinematic parameters. 

Moreover, the similarities between the Pap and the Centr_Pap regarding the motility and the kinematic parameters (Table 4 and Appendix A) demonstrate the complete degradation of the macro and microviscosity of the seminal plasma, performed by the Papain enzyme. The movement of spermatozoa through the TCF-ejaculate column during centrifugation could have been improved by the liquefaction of the ejaculates, resulting in higher sperm recovery rates. Regarding this aspect, it would be very interesting to assess if the enzymatic reduction of viscosity would allow for a higher recovery rate following a single-layer centrifugation procedure, which currently is reported to be around 30% [13].

The positive effects of Papain treatment upon freezing have been previously reported both in the alpaca [22] and in the white rhinoceros (*Ceratotherium simum*) [35] but mainly post-thawing. In those studies, Papain was found to be non-detrimental to the membrane or acrosome status of spermatozoa and indeed beneficial for the cryosurvival of spermatozoa [22,35]. With that in mind, the results of the present study should be interpreted with caution until the effects of such treatments can be evaluated, particularly after semen preservation (chilling, freezing). 

In the present study, differences in agglutination scores were observed amongst ejaculates, regardless of the addition of Papain. However, the addition of D-penicillamine to the TCF extender had no effect on the incidence of DC sperm head-to-head agglutination despite previously being reported as effective at preventing this phenomenon in the ram [23]. Though not significantly, papain treatment increased head-to-head agglutination, as had also been observed before [20], but only if it was already present in the pre-treatment ejaculates. In order to better understand this aspect and aspects related to the SP, samples of Exp 1 and Exp 3 were fixed, smeared, and stored at 4 °C for additional studies to be performed on sperms’ lectins and glycocalyx. Those studies will also include the evaluation of the PNA reactive sperms (acrosomes) [24], but due to the high number of lectins to be evaluated, this analysis is expected to be completed at a later date and thus will be reported and discussed in another study. 

## 5. Conclusions

The present study provides further insights into DC semen processing procedures and opens up a new area of possible investigations related to the effects of seminal plasma on DC spermatozoa. Moreover, the results of this investigation show that a simple, cheap, and effective procedure, such as cushioned centrifugation, could improve the motility patterns of dromedary camel spermatozoa. In combination with enzymatic reduction of viscosity, this method leads to the best results in terms of recovery rates and kinematic parameter. However, further studies are needed in order to find the best semen processing protocol for this species. Future work may benefit from sperm motility and glycocalyx evaluations and must ultimately be validated by the achievement of satisfactory pregnancy rates following artificial insemination. 

## Figures and Tables

**Figure 1 animals-13-02685-f001:**
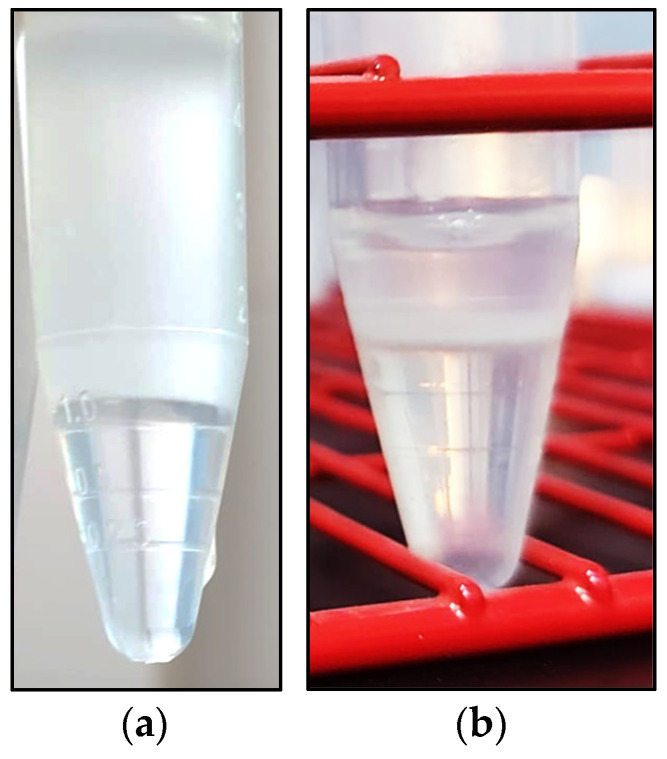
Cushion fluid layered below a TCF diluted dromedary camel ejaculate (**a**) and sperm buffer (arrow) following centrifugation at 900× *g* × 20 min at RT (**b**).

**Figure 2 animals-13-02685-f002:**
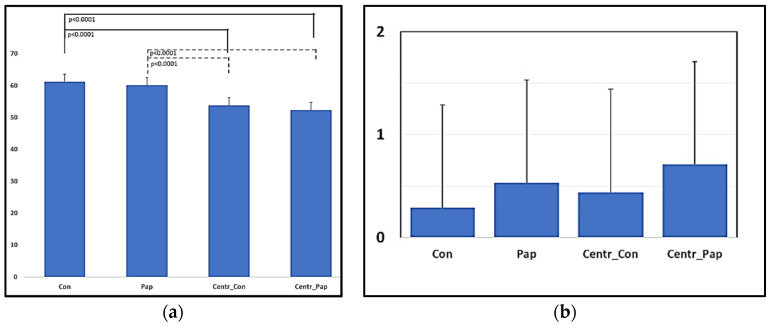
Effects of Papain 0.1 mg/mL (5 min) + E64 10 µM and cushioned centrifugation (900× *g* × 20 min at RT) on the viability (**a**) (%) and agglutination score (**b**) (0–5) of dromedary camel diluted ejaculates.

**Table 1 animals-13-02685-t001:** Effects of 35% Seminal Plasma incubation (45 min at 37 °C) on the motility and viability percentages of dromedary camel bull epididymal spermatozoa.

	Control	Control Time	Seminal Plasma	F-Test	*p*-Value	Control vs. Con_Time	Control vs. S_Plasma	S_Plasma vs. Con_Time
	Mean ± St. Dev.	Mean ± St. Dev.	Mean ± St. Dev.					
Motile	69.99 ± 18.85	70.62 ± 17.67	64.62 ± 22.29	3.22	0.096	n.s.	n.s.	n.s.
Progr Motile	60.21 ± 21.77	59.26 ± 24.22	37.36 ± 22.01	17.41	**<0.0001**	1	**0.0047**	**0.0149**
Rapid	5.04 ± 3.08	4.38 ± 3.33	2.65 ± 1.94	2.74	0.1	n.s.	n.s.	n.s.
Medium	55.17 ± 22.76	54.87 ± 22.55	34.64 ± 21.42	15.85	**0.002**	1	**0.0064**	**0.0017**
Non Progressive	9.78 ± 4.58	11.37 ± 6.84	27.40 ± 10.03	13.95	**0.005**	1	**0.005**	**0.041**
Immotile	30.01 ± 18.85	29.38 ± 17.67	35.31 ± 21.99	3.18	0.098	n.s.	n.s.	n.s.
Live	88.50 ± 4.71	84.44 ± 5.35	85.03 ± 2.98	10.66	**0.003**	**0.0077**	**0.0095**	1

Control: before incubation; Control Time: after incubation; Seminal Plasma: after 45 min incubation (35% Seminal Plasma).

**Table 2 animals-13-02685-t002:** Effects of centrifugation (900× *g* × 20 min at RT), with and without cushion fluid, on the motility, viability, and intact plasma membrane (HOST) percentages of dromedary camel bull spermatozoa.

	Pre_Centr	Centr_Con	Centr_Cush	*p*-Values	Pre_Centr vs. Centr_Con	Pre_Centr vs. Centr_Cush	Centr_Con vs. Centr_Cush
	Mean ± St. Dev.	Mean ± St. Dev.	Mean ± St. Dev.				
Motile	58.66 ± 8.72	62.09 ± 11.18	63.98 ± 11.25	0.123	0.631	1	0.1953
Progr Motile	43.53 ± 13.04	43.76 ± 19.96	53.26 ± 15.89	**0.002**	1	**0.009**	**0.0022**
Rapid	1.51 ± 1.06	1.49 ± 2.01	1.77 ± 1.81	0.704	1	0.9024	1
Medium	42.03 ± 13.17	42.3 ± 19.49	51.48 ± 15.89	**0.002**	1	**0.0081**	**0.0015**
Non Progressive	15.93 ± 8.06	18.26 ± 13.17	10.61 ± 6.63	**0.014**	0.9947	0.0937	**0.0355**
Immotile	41.06 ± 8.51	37.94 ± 11.08	36.13 ± 11.39	0.191	0.8508	1	0.3151
Live	75.73 ± 7.12	66.16 ± 9.42	68.55 ± 10.55	**<0.001**	**0.0002**	**0.0086**	0.2096
HOST	64.62 ± 10.04	53.24 ± 8.52	55.66 ± 5.49	**0.001**	**0.0192**	**0.0327**	0.3383

Pre_Centr: Before centrifugation; Centr_Control: Centrifugation without Cushion; Centr_Cush: Centrifugation above 1 mL of Cushion fluid.

**Table 3 animals-13-02685-t003:** Effects of centrifugation (900× *g* × 20 min at RT), with and without cushion fluid, on kinematic parameters of dromedary camel bull spermatozoa.

		Pre_Centr	Centr_Con	Centr_Cush	Pre_Centr vs. Centr_Con	Pre_Centr vs. Centr_Cush	Centr_Con vs. Centr_Cush
		Mean ± S.E.	Mean ± S.E.	Mean ± S.E.			
VCL (µm/s)	Motile	76.43 ± 10.56	92.58 ± 11.45	110.6 ± 12.56	0.0859	**<0.0001**	**0.0013**
	Progr Motile	83.07 ± 10.77	106.42 ± 10.31	119.24 ± 11.63	0.6576	**0.0007**	0.5382
VAP (µm/s)	Motile	32.74 ± 4.44	38.04 ± 4.17	44.76 ± 4.66	0.9376	**<0.0001**	**0.0263**
	Progr Motile	37.38 ± 3.97	43.37 ± 3.73	48.86 ± 4.34	0.9103	**0.0045**	0.3189
VSL (µm/s)	Motile	15.73 ± 2.15	18.73 ± 1.87	21.94 ± 1.80	0.9233	**<0.0001**	0.2596
	Progr Motile	17.68 ± 2.19	21.42 ± 1.70	24.09 ± 1.65	0.8531	**<0.0001**	0.5931
STR (%)	Motile	46.88 ± 1.32	49.93 ± 1.22	49.95 ± 1.88	0.4289	0.7746	1
	Progr Motile	47.29 ± 1.56	50.38 ± 1.23	50.78 ± 2.01	0.595	0.6658	1
LIN (%)	Motile	21.36 ± 0.70	21.96 ± 0.81	21.24 ± 1.04	1	1	0.9998
	Progr Motile	21.59 ± 0.57	21.84 ± 0.61	21.70 ± 1.06	1	1	1
WOB (%)	Motile	45.12 ± 0.79	43.85 ± 1.10	42.61 ± 0.79	0.9996	0.7754	0.8817
	Progr Motile	45.67 ± 0.62	43.22 ± 0.94	42.78 ± 0.77	0.8119	0.4008	1
ALH (µm)	Motile	2.07 ± 0.26	2.54 ± 0.29	2.93 ± 0.32	0.7809	**0.001**	**0.0305**
	Progr Motile	2.31 ± 0.24	2.87 ± 0.27	3.16 ± 0.30	0.6013	**0.0054**	0.4305
BCF (Hz)	Motile	8.68 ± 0.65	9.19 ± 0.49	10.41 ± 0.45	0.9993	**<0.0001**	**0.002**
	Progr Motile	10.19 ± 0.41	10.69 ± 0.34	11.49 ± 0.29	0.9991	**<0.0001**	0.3421

VCL: curvilinear velocity; VAP: average path velocity; VSL: straight line velocity; STR: straightness; LIN: linearity; WOB: wobble; ALH: amplitude of lateral head displacement; BCF: beat cross frequency. Pre_Centr: Before centrifugation; Centr_Control: Centrifugation without Cushion; Centr_Cush: Centrifugation above 1 mL of Cushion fluid.

**Table 4 animals-13-02685-t004:** Effects of Papain 0.1 mg/mL (5 min) + E64 10µM, followed by cushioned centrifugation (900× *g* × 20 min at RT), on the motility percentages of dromedary dromedary camel diluted ejaculate.

	Con	Pap	Centr_Con	Centr_Pap	Con vs. Pap	Con vs. Centr_Con	Con vs. Centr_Pap	Centr_Con vs. Centr_Pap	Pap vs. Centr_Pap
	Mean ± St. Dev.	Mean ± St. Dev.	Mean ± St. Dev.	Mean ± St. Dev.					
Motile	33.66 ± 3.03	41.72 ± 3.86	32.63 ± 3.42	41.39 ± 3.31	0.31	1.00	**0.01**	**0.0249**	1
Progr Motile	12.54 ± 2.59	29.53 ± 4.81	17.67 ± 2.99	29.80 ± 4.20	**0.0097**	0.5065	**0.0001**	**0.0023**	1
Rapid	1.77 ± 0.49	2.62 ± 0.52	2.33 ± 0.58	3.58 ± 0.90	0.9684	0.3454	0.8332	0.998	0.9951
Medium	10.77 ± 2.37	26.91 ± 4.65	15.37 ± 2.76	26.22 ± 4.22	**0.0099**	0.6598	**0.0013**	**0.0036**	1
Non Progr	21.12 ± 2.16	12.18 ± 2.41	14.94 ± 2.09	11.59 ± 2.10	**0.0255**	**0.0239**	**0.0159**	0.9648	1
Immotile	66.34 ± 3.03	58.28 ± 3.86	67.37 ± 3.42	58.61 ± 3.31	0.3124	1	**0.0114**	**0.0247**	1

Con: Control before centrifugation; Pap: before centrifugation after treatment with Papain + E64: Centr_Con: Control after centrifugation; Centr_Pap: after treatment with Papain + E64 and centrifugation.

**Table 5 animals-13-02685-t005:** Effects of enzymatic reduction of seminal plasma viscosity with Papain 0.1 mg/mL (5 min) + E64 10 µM, before and after a cushioned centrifugation procedure (900× *g* × 20 min at RT), on kinematic parameters of dromedary camel spermatozoa.

		Con	Pap	Centr_Con	Centr_Pap	Con vs. Pap	Centr_Convs Centr_Pap	Con vs. Centr_Con
		Mean ± S.E.	Mean ± S.E.	Mean ± S.E.	Mean ± S.E.			
VCL (µm/s)	Motile	66.82 ± 5.08	120.63 ± 12.23	86.08 ± 6.00	120.33 ± 10.16	**<0.0001**	**0.0011**	**0.0043**
	Progr Motile	103.78 ± 3.98	142.51 ± 9.02	120.55 ± 4.32	141.30 ± 7.56	**0.0002**	**0.0048**	**0.001**
VAP (µm/s)	Motile	31.10 ± 2.33	52.75 ± 5.35	39.40 ± 3.02	51.22 ± 4.04	**<0.0001**	**0.0004**	**0.0064**
	Progr Motile	47.57 ± 2.15	61.85 ± 4.00	53.75 ± 2.67	60.13 ± 2.99	**0.0006**	0.0898	0.0728
VSL (µm/s)	Motile	14.64 ± 1.37	22.98 ± 2.28	18.96 ± 1.67	24.10 ± 1.92	**0.0001**	**0.0062**	**<0.0001**
	Progr Motile	23.80 ± 1.54	28.26 ± 1.80	26.37 ± 1.97	29.48 ± 1.72	**0.0303**	0.8115	0.9682
STR (%)	Motile	45.53 ± 1.54	43.49 ± 1.67	46.26 ± 1.49	46.73 ± 1.94	0.9903	1	1
	Progr Motile	49.29 ± 2.05	47.21 ± 2.17	49.37 ± 2.21	50.23 ± 2.50	1	1	1
LIN (%)	Motile	21.29 ± 0.94	19.22 ± 0.82	21.75 ± 0.83	20.36 ± 0.73	0.7859	0.9932	1
	Progr Motile	23.66 ± 1.27	20.91 ± 0.98	22.87 ± 1.35	21.77 ± 0.96	0.8767	1	1
WOB (%)	Motile	46.97 ± 1.34	44.17 ± 1.16	45.94 ± 1.04	43.86 ± 1.06	0.7644	0.9375	0.9919
	Progr Motile	46.59 ± 1.49	44.05 ± 1.00	45.37 ± 1.30	43.55 ± 0.99	0.812	0.9856	0.9997
ALH (µm)	Motile	1.92 ± 0.12	3.20 ± 0.29	2.39 ± 0.15	3.18 ± 0.25	**0.0001**	**0.0015**	**0.0137**
	Progr Motile	2.81 ± 0.10	3.72 ± 0.21	3.22 ± 0.11	3.68 ± 0.19	**0.0002**	**0.019**	**0.001**
BCF (Hz)	Motile	8.94 ± 0.51	11.63 ± 0.75	9.78 ± 0.55	11.63 ± 0.48	**<0.0001**	**<0.0001**	0.4974
	Progr Motile	12.04 ± 0.46	13.55 ± 0.56	12.18 ± 0.47	13.62 ± 0.39	0.3199	**0.0001**	1

VCL: curvilinear velocity; VAP: average path velocity; VSL: straight line velocity; STR: straightness; LIN: linearity; WOB: wobble; ALH: amplitude of lateral head displacement; BCF: beat cross frequency. Con: Control before centrifugation; Pap: before centrifugation after treatment with Papain + E64: Centr_Con: Control after centrifugation; Centr_Pap: after treatment with Papain + E64 and centrifugation.

## Data Availability

Data could be provided upon email request to the corresponding author.

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
