# Peer review of "The Effect of Cushioned Centrifugation, with and without Enzymatic Reduction of Viscosity, on the Motility Pattern and Kinematic Parameters of Dromedary Camel Bull Spermatozoa"

_animals, 2023, doi:10.3390/ani13172685_

Round 1

Reviewer 1 Report

This study investigated the cushioned centrifugation together with enzymatic reduction of viscosity on camelids sperm quality, especially in terms of motion parameters. The attempts in this area will benefit the development of semen handling procedure in camelids. The reviewer provides the following comments with an aim of helping to improve the presentation of this study.

 Line 24, 130, why did the authors choose 35% SP, not in other percentage, in this study?

Line 33, what is “medium spermatozoa”?

Line 36, the “recovery rates” was not reflected in the results, it is recommended to change this phrase.

Line 15, 25, there is a confusion with the sperm samples, in Simple Summary it says “ejaculates” while in Abstract only epididymal sperm are mentioned in the first objective of the study. All the information should be in consistence with each other and sperm samples in each part should be clearly stated.

Line 55, the extender used by other researchers should be given here, in order to give a comparison with the following citation.

Line 63, the authors are suggested to describe what is cushioned centrifugation and make comparisons with the normal centrifugation in terms of set parameters or equipment needed.

Line 74, too many abbreviations, like DC here, make the reading difficult.

Line 108-112, it is not clear how seminal plasma samples were prepared. Were they from diluted ejaculates?

Line 128-134, the authors are suggested to put this information in experiment design so that the design is easier to understand and the methods are clearly depicted.

Line 136, the detailed information of CASA system should be given here. In addition, it is necessary to describe how the kinematic parameters were obtained.

In Materials and Methods, it is suggested to reconstruct the content by putting all the information of experiment design and specific measurements separately, as there are too many content repeated in each experiment, e.g. quality evaluations.

Line 187, Figure 1 and 2, it is better to put it as Figure 1 only and mark each picture before and after centrifugation with arrows pointing the layer separated.

 Line 206, the authors should state the reason why you chosen the concentration of Papain as 0.1 mg/mL.

 Line 232, concerning to the statistics, it should be concise and brief describing the methods you used in this study.

Line 278, it is recommended to add a title to each part of the results.

The tables and figures in this study should be presented formally. The current form is a little bit hard to understand.

The authors are recommended to make a sufficient discussion on the underlying reason why the cushioned centrifugation combined with the viscosity reduction enzyme had a better effect on sperm parameters.

minor modification

Author Response

This study investigated the cushioned centrifugation together with enzymatic reduction of viscosity on camelids sperm quality, especially in terms of motion parameters. The attempts in this area will benefit the development of semen handling procedure in camelids. The reviewer provides the following comments with an aim of helping to improve the presentation of this study.

 Line 24, 130, why did the authors choose 35% SP, not in other percentage, in this study?

RE: The 35% SP concentration was chosen cause diluting in 100% SP would have required the centrifugation of the Epididymal semen (after recovering with TCF) and thus an additional factor that might have affected the sperm’s motility. In the discussion section also it is reported: “the addition of TCF was a necessary step for decreasing ejaculate viscosity during storage at 4°C. Processing of ejaculates at reduced temperatures was performed in order to reduce the metabolic activity of spermatozoa (during liquefaction and centrifugation); this pre-caution could possibly have prevented the observation of any negative effects resulting from altered seminal plasma”

Line 33, what is “medium spermatozoa”?

RE: The definisition of medium spematozoa is reported at line 198 “Medium spermatozoa >30 µm/s <50 µm/s;”. We added the word “progressive” within the text.

Line 36, the “recovery rates” was not reflected in the results, it is recommended to change this phrase.

RE: The recovery rates results were reported in lines 326-328 for experiment 2 “The recovery rates for normal and cushion fluid centrifugation were 78.2±8.3% and 79.6±13.4% respectively, and no statistical difference were found among the two proce-dures (p=0.7896)”

and in lines 387-389 for experiment 3 “The median recovery rate of sperm for the Centr_Pap samples was significantly higher (p<0.0001) than the Centr_Con: 86.37% (Range 74.06-92.84) vs 78.16% (Range 56.09-90.32), respectively.”

Line 15, 25, there is a confusion with the sperm samples, in Simple Summary it says “ejaculates” while in Abstract only epididymal sperm are mentioned in the first objective of the study. All the information should be in consistence with each other and sperm samples in each part should be clearly stated.

RE: The term “epididymal” has been added at line 16 and the term “spermatozoa” has been replaced with “ejaculates” at line 27.

Line 55, the extender used by other researchers should be given here, in order to give a comparison with the following citation.

RE: The sentence has been modified LINE 57-58: “For instance, in the dromedary camel (DC) species some authors   diluted ejaculates (1:1, v/v) with an commercial extenders (Green Buffer, Optixcell, Triladyl)”

Line 63, the authors are suggested to describe what is cushioned centrifugation and make comparisons with the normal centrifugation in terms of set parameters or equipment needed.

RE: There are different time and force used for different species: Authors do not consider useful to report different centrifugation regimes also in view of the different protocols proposed  for each species.

Lines 68-70: The sentence was modified in order to describe the cushion fluid: “The cushion fluid is a non-ionic iodinated compound, “iodixanol” wich reduces the pressure that the spermatozoa undergo during centrifugation [19]. Thank You for the observation: indeed we did not specified that a swing rotor centrifuge was used, hence we added this information at Lines 186 and 231-232. “with a swing rotor centrifuge (Hermle, Z326K, Germany)”

Line 74, too many abbreviations, like DC here, make the reading difficult.

RE: We have used abbreviations, according to standard manuscript writing guidelines in order to made the manuscript more readable. We have corrected the sentence at line 80 and throughout the manuscript; please suggest where other corrections might be needed.

Line 108-112, it is not clear how seminal plasma samples were prepared. Were they from diluted ejaculates?

RE: The seminal plasma preparation has been described in lines 100-114: “The selected ejaculates (n° 4, non-contaminated, volume ≥4 mL, sperm concentration >300 x 106 sperms/mL and total motility >80%) were added with 30% (v/v) TCF. Diluted ejaculates were kept at 4°C, pipetted intermittently, and allowed to liquefy at 4°C. Following complete liquefaction, samples were centrifuged twice at 1,000 x g for 30 min at 4°C or until the recovered supernatant was completely free of spermatozoa. Seminal plasma samples were pooled and stored in aliquots at -80°C until use.

Line 128-134, the authors are suggested to put this information in experiment design so that the design is easier to understand and the methods are clearly depicted.

RE: we have included the requested information in lines 95-102 “selected samples (concentration of >300 x 106 spz/mL, total motility > of 75%) were pooled, samples were diluted 1:1 with 70% SP and evaluated after 45 minutes incubation at 37°C.” and Lines 100-101: “(non-contaminated ejaculates with a concentration > 250 x 106 spz/mL, mass motility score ≥3)”.

Line 136, the detailed information of CASA system should be given here. In addition, it is necessary to describe how the kinematic parameters were obtained.

RE: Unfortunately, we had several complains about the company which provided the CASA system (warrancy and assistance). We have decided not to report the information about the system for not advertising the instrument or the company. Since the setting of the instrument is reported (lines 144-146) the experiment and the results are repeatable.

What exactly the reviewer means with “describe how the kinematic parameters were obtained”?

In Materials and Methods, it is suggested to reconstruct the content by putting all the information of experiment design and specific measurements separately, as there are too many content repeated in each experiment, e.g. quality evaluations.

RE: We have indicated the procedure of kinematic parameter evaluation only for experiment 1 and referred to that paragraph for Exp 2 and 3.

Line 187, Figure 1 and 2, it is better to put it as Figure 1 only and mark each picture before and after centrifugation with arrows pointing the layer separated.

RE: done

Line 206, the authors should state the reason why you chosen the concentration of Papain as 0.1 mg/mL.

RE: the 0.1 mg/mL concentration was chosen according to reference n°22 (added to the sentence).

Line 232, concerning to the statistics, it should be concise and brief describing the methods you used in this study.

RE: The paragraph about statistics has been shortened: Lines 240-268.

Line 278, it is recommended to add a title to each part of the results.

RE: Done

The tables and figures in this study should be presented formally. The current form is a little bit hard to understand.

RE: Tables and figures were provided in separate files but they were included in the manuscript, by the editorial office, without formatting. We have modified the tables within the manuscript and we hope that data could now be evaluated in a better way.

The authors are recommended to make a sufficient discussion on the underlying reason why the cushioned centrifugation combined with the viscosity reduction enzyme had a better effect on sperm parameters.

RE: Thank You for this important suggestion, we have added a small paragraph (lines 502-507) in order to elucidate this aspect “Moreover, the similarities between the Pap and the Centr_Pap regarding the motility and the kinematic parameters (Table 4 and Supplementary Table 3) demonstrates the complete degradation of the macro and microviscosity of the seminal plasma, performed by the Papain enzyme. The movement of spermatozoa through the TCF-ejaculate column, during centrifugation, could have been improved by the liquefaction of the ejaculates, resulting in higher sperm’s recovery rates.”

Reviewer 2 Report

In general, the material and methods section provides a detailed description of the experimental procedures carried out in the study. It includes information on the site, ethical approval, experimental design, semen collection and evaluation methods, dilution techniques and statistical analysis. However, the section lacks the specific criteria used for the selection of samples for the experiments. As well as the number of samples, although it is reflected in the Results section in experiments 2 and 3, none of the experiments reflect the number of animals from which the testes (exp 1) or ejaculates (exp 2 and 3) were derived. This should be added well defined in the material and methods section.

Discussion:

The provided discussion section corresponds to the results of the study and is scientifically well defended. It begins by addressing the background and gaps in knowledge regarding assisted reproductive technologies and the effects of seminal plasma on dromedary camel spermatozoa. The discussion then presents the findings of the study, comparing them with previous research in alpacas and other species. Overall, the discussion provides a comprehensive analysis of the study's findings, compares them with previous research, and highlights the potential implications and areas for further investigation.

Author Response

Reviewer II

In general, the material and methods section provides a detailed description of the experimental procedures carried out in the study. It includes information on the site, ethical approval, experimental design, semen collection and evaluation methods, dilution techniques and statistical analysis. However, the section lacks the specific criteria used for the selection of samples for the experiments. As well as the number of samples, although it is reflected in the Results section in experiments 2 and 3, none of the experiments reflect the number of animals from which the testes (exp 1) or ejaculates (exp 2 and 3) were derived. This should be added well defined in the material and methods section.

Discussion:

The provided discussion section corresponds to the results of the study and is scientifically well defended. It begins by addressing the background and gaps in knowledge regarding assisted reproductive technologies and the effects of seminal plasma on dromedary camel spermatozoa. The discussion then presents the findings of the study, comparing them with previous research in alpacas and other species. Overall, the discussion provides a comprehensive analysis of the study's findings, compares them with previous research, and highlights the potential implications and areas for further investigation

RE: We thank the reviewer for their approval of our work

RE: The ejaculates selection criteria were indicated in the Materials and Methods; they were also indicated in the Experimental design paragraph (lines 128-134) according to the request of the reviewer I.

In Experiment 1 semen samples were obtained by pooling epididymal spermatozoa obtained from a MINIMUM of 3 epididymal semen samples (Lines 134-135). Time to time, pooling was performed according to the testis obtained from the slaughterhouse: sometimes 4 testis, sometimes 3; the number of males from which testis were collected is, therefore, not relevant.

In experiments 2 and 3 samples were obtained by 5 different bulls; the statistical analysis did not find any influence of the bull on the obtained results. Since there are no reports in literature about the individual bulls’ (or stallion) effect on the motility of centrifuged spermatozoa (as compared with post-thaw motility, where good or bad freezers bulls have been identified) this information was omitted.

Round 2

Reviewer 1 Report

After a careful revison, this manuscript has been improved. The authors have responded the comments and suggestions, and appropriate modifications have been made. No further comments.

minor

Author Response

Thank You again for your revision.

The manuscript has been revised by a native English speaker.

We kindly ask the reviewer to indicate if and where further revision are requested.

Thank You in advance

Davide Monaco